# Multidisciplinary Approach to Characterizing the Fingerprint of Italian EVOO

**DOI:** 10.3390/molecules24081457

**Published:** 2019-04-12

**Authors:** Marco Abbatangelo, Estefanía Núñez-Carmona, Giorgio Duina, Veronica Sberveglieri

**Affiliations:** 1Department of Information Engineering, University of Brescia, Brescia, via Branze, 38, 25123 Brescia, BS, Italy; giorgio.duina@unibs.it; 2CNR-IBBR, Institute of Bioscience and Bioresources, via Madonna del Piano, 10, 50019 Sesto Fiorentino, FI, Italy; estefania.nunezcarmona@ibbr.cnr.it (E.N.-C.); veronica.sberveglieri@ibbr.cnr.it (V.S.); 3NANO SENSOR SYSTEMS, NASYS Spin-Off University of Brescia, Brescia, via Camillo Brozzoni, 9, 25125 Brescia, BS, Italy

**Keywords:** sensors array, nanowire gas sensors, EVOO, GC-MS-SPME, PCA, k-NN

## Abstract

Extra virgin olive oil (EVOO) is characterized by its aroma and other sensory attributes. These are determined by the geographical origin of the oil, extraction process, place of cultivation, soil, tree varieties, and storage conditions. In the present work, an array of metal oxide gas sensors (called S3), in combination with the SPME-GC-MS technique, was applied to the discrimination of different types of olive oil (phase 1) and to the identification of four varieties of Garda PDO extra virgin olive oils coming from west and east shores of Lake Garda (phase 2). The chemical analysis method involving SPME-GC-MS provided a complete volatile component of the extra virgin olive oils that was used to relate to the S3 multisensory responses. Furthermore, principal component analysis (PCA) and k-Nearest Neighbors (k-NN) analysis were carried out on the set of data acquired from the sensor array to determine the best sensors for these tasks and to assess the capability of the system to identify various olive oil samples. k-NN classification rates were found to be 94.3% and 94.7% in the two phases, respectively. These first results are encouraging and show a good capability of the S3 instrument to distinguish different oil samples.

## 1. Introduction

Oil is the product obtained from the milling of olive drupes (*European Olea*), from a fruit tree belonging to family Oleaceae (*Dicotyledons*). Olive tree is one of the oldest cultivated plants in the world. Its fruit is a drupe weighing between 0.5 and 20 g, formed by the epicarp, mesocarp, endocarp and, more internally, the seed. The components that are quantitatively more important than fresh fruit include water (40–70%) and fat substances (6–25%), which are contained mainly in the mesocarp. The oil is mainly present in the pulp (16–25% of the fresh weight) and is limited in the drupe almond of the olive (1–1.5% of the fresh weight) [1]. The entire technological process of oil production from olives provides for the exclusive use of physical means and mechanics in the various phases leading to the separation of the oily component from the pomace and water of vegetation. 

From a chemical-physical point of view, olive oil can be considered a liquid food fat at room temperature. The chemical composition of olive oil can be very simply schematized from the point of view of the respective proportions, in major and minor components. The series of major compounds (98–99%) is made up mainly of triacylglycerols or triglycerides (TAG) and from the group consisting of free fatty acids (FFA) and diglycerides (MAG and DAG). The fatty acids that make up triglycerides are mainly palmitic C16: 0 (P: 6.30–20.93%), palmitoleic C16: 1 (Po: 0.32–3.52%), stearic C18: 0 (S: 0.32–5.33%), oleic C18: 1 (O: 55.23–86.64%), linoleic C18: 2 (L: 2.7–20.24%) and linolenic C18: 3 (Ln: 0.11–1.52%) [2]. Other derivatives of fatty acids, such as phospholipids, waxes and esters of the sterols, are traditionally included in the minor compounds, together with a large and heterogeneous group of over 200 compounds consisting of sterols, aliphatic alcohols, hydrocarbons, biophenols, tocopherols, volatile compounds. Although quantitatively they are small concentrations, at most a few hundred ppm, such minor compounds play an important role in defining the specific characteristics of the uniqueness of olive oil as an alimentary fat both from the point of view of sensory and nutritional quality, in the definition of typology and genuineness, and, more recently, in the field of traceability, in addition to health aspects. All these compounds underlie current and future challenges in the field of scientific research on EVOO [3].

EVOO is obtained through purely physical and mechanical processes. This makes it a product with particular characteristics, especially from the point of view of organoleptic and nutritional properties, which play a fundamental role in the eating habits of Mediterranean populations. Volatile compounds derive from the natural biochemical reactions that are triggered in the olive during the milling and the extraction of the oil in the crusher. Their composition is at the base of the diversity of aroma that is found between individual olive oils and varies with the variety of olive, with the degree of ripeness, with the extraction temperature, and with various other factors, contributing to the typology of the product. It has been widely demonstrated in the literature that cultivars, climatic conditions of the place of cultivation, agronomic practices, degree of maturation, storage conditions and fruit processing techniques are all factors that can influence the formation and presence of compounds responsible for the aroma of virgin oil olive, both in the typology and in the quantity of volatile and phenolic compounds. Olives of the same variety, cultivated in the same environmental conditions, produce oils with different volatile compounds, as also happens for the same cultivar when grown in different areas [4]. 

These typical features are of great interest to the customers, since they are linked to the olive variety or cultivars from which the EVOO is elaborated, and to their specific geographical origin or Protected Designation of Origin (PDO) know-how applied for their production [5,6]. Thus, the composition and organoleptic characteristics of EVOO are strongly correlated with the geographical origin of production. From this perspective, great efforts have been made to catalogue and characterize the sensory and volatile profile of monovarietal EVOOs produced in a delimited geographical area. Moreover, the assessment of quality and geographical origin of EVOOs is becoming more and more important from the perspective of both the olive oil manufacturers and the customers. A differentiation based on the geographical or varietal origin and high-quality standards is one of the most powerful competitive strategies that olive oil producers are currently adopting to face the challenges of fraudulent activities and adulteration of EVOOs.

The necessity of developing accurate analytical methods has prompted the present study to attempt the use of an array of metal oxide (MOX) gas sensors to discriminate the geographical origin of different PDO EVOOs. Gas sensors are a typology of chemical sensors that transduce a chemical interaction in an electrical signal. Interactions happen between volatile compounds and sensing material; this causes an exchange of electrons and a variation of conductivity of the sensor. MOX gas sensors are non-specific sensors; hence, they respond to different classes of compounds with different sensibility. For this reason, in order to understand which compounds have been detected by sensors, they are “trained” on a consistent dataset. Furthermore, this is the main reason that sensor arrays are used in most applications. In recent years, electronic nose systems with MOX semiconductor gas sensors have received much attention in the literature for the determination or classification of the geographical provenance of EVOOs [7,8,9,10,11,12]. The link between an EVOO and the geographical area can characterize the product itself and determine its nutritional and organoleptic characteristics. Furthermore, the volatile composition of an EVOO is directly related to the production area, quality categories, and cultivars. All these factors interact with each other, resulting in a complex multivariate matrix [13]. 

In this study, the SPME-GC-MS, conducted in parallel with a versatile MOX nanowires gas sensor-based electronic nose (called S3), was applied to characterize the volatile chemical profile of EVOOs in relation to their geographical origins. The novelty of this approach compared to the cited papers lies in the use of sensors with nanotechnological properties. The aims of this work were twofold: firstly, it was important to assess the capability of the system to distinguish between olive oil, non-PDO EVOOs and PDO EVOOs; secondly, attention was focused on the discrimination of the four varieties of Garda PDO EVOOs, two from the eastern side of the lake, and the others from the western side.

## 2. Results and Discussion

### 2.1. Sensor Responses

Once the data were acquired, sensor responses were checked for the measurements of each session. Moreover, the response curves for the given olive oil samples obtained from the individual sensor were studied to observe and detect any relevant features. This step allowed us to eliminate the sensors whose response did not yield useful information. In Figure 1, it is possible to see a typical response of a n-type MOX sensor; in this case, normalized resistance variation of a SnO2+Au nanowire is shown. Normalization of the signals has been performed computing the ratio between the first value of resistance recorded and all the values of the same measure. For this reason, all signals start from a value equal to 1. Sensors responses are grouped based on their belonging to one of the 4 samples analyzed in the first phase: olive oil, non-PDO EVOOs and Garda PDO EVOO. Furthermore, signals are shown in sequence along the time axis; since each measurement lasts 600 s, time ranges from 0 s to 2400 s.

All the curves present a decreasing value of resistance during VOC analysis, while it reaches the baseline when filtered ambient air is fluxed inside the sensor chamber. It is evident from the graph that the three categories of analyzed olive oil differ in terms of the minimum value of resistance; this value is reached during the VOC exposition. The olive oil (in blue) is the one with the minimum variation, while non-PDO EVOOs in green and orange reached similar values, and all of them differ from the Garda PDO sample. Since the first goal was the discrimination between olive oil samples, non-PDO EVOOs and PDO EVOOs, the observed normalized resistance variation was considered as the first feature to be extracted from data. 

Then, the first derivative of the signal over time was considered. The first derivative can give information regarding the speed at which the resistance varies over time, which is an index of the speed of the reaction that take place over sensing layers. Indeed, since different oils are characterized by different numbers and amounts of VOCs, and sensors have different affinity toward them, it is plausible to think that the maximum speed of resistance change can be used as a marker for different oil typologies. In Figure 2, the first derivatives of SnO_2_ RGTO sensor are shown. The sequence representation of the signals follows the same logic described for Figure 1. In this case, the maximum speed is the minimum (negative) value, since derivatives were calculated using finite differences, a method widely used to approximate numerical derivatives.

It is clear that this RGTO sensor reaches the minimum value of resistance at different speeds, depending on the oil under analysis. The olive oil (blue) reaches the lowest negative value, while the Garda PDO EVOO reaches the highest value. In the middle, there are two non-PDO EVOO samples; unlike what was seen in Figure 1, in this case, there is a clear difference between them. Even though this is not important for the designated aim, it could be very useful in future analyses in cases where it is necessary for a device to be able to recognize different non-EVOOs.

To evaluate which of the 14 features (7 sensors × 2 types of features) were suitable for the task, Principal Component Analysis (PCA) was performed. In Figure 3, score (top) and loading (bottom) plots are shown; the total explained variance was equal to 88.55%. A clear distinction between the olive oil sample (red circles) and the EVOOs can be noticed. On the contrary, there is a partial overlap between the Garda PDO (blue cluster) and the other two EVOOs (in green). This result could mainly be due to different production processes that distinguish a regular olive oil from an extra virgin one, but also to the different cultivar composition of the analyzed samples. Consequently, the number and amount of the VOCs differ from sample to sample, leading to dissimilar responses of the sensors. However, the use of sensors that respond in a similar way are redundant, since they provide the same information. For this reason, some of them are discarded. Dimension reduction of the dataset was done through the analysis of loadings. Starting from 14 parameters, 7 were not considered for the following analysis, since they gave the same contribution to the first two principal components (PCs). The remaining features were ΔR/R0 values of two SnO_2_-RGTO, SnO_2_Au-RGTO, CuO, SnO_2_Au-NW and the minimum of 1st derivative of SnO_2_-RGTO and SnO_2_Au+Au-NW.

Selected parameters were used as input for k-Nearest Neighbors (k-NN) classification algorithm. It was found that the optimal k = 4, with a classification rate equal to 94.3%. Three Garda PDO samples were confused with non-PDO EVOOs, and one olive oil specimen was misclassified as PDO EVOO. All the samples of non-PDO EVOOs were correctly identified.

Once the system was tested and we had noted that it was able to distinguish three different types of oil with a good accuracy, we tried to evaluate whether S3 was able to distinguish among two different PDO EVOOs. Four different Garda products were considered; from here on, they are called Western Garda and Eastern Garda. The same steps were used as in the first phase of data analysis. PCA in Figure 4 shows the separation between the two different lake shore EVOOs (red and black clusters), while the other two seem to be more similar, since they are partially overlapped (blue and green groups). The total explained variance is equal to 80.65% in the first two PCs.

Also, in this case, the loading analysis led to a selection of 7 parameters out of the 20 used (10 sensors x 2 features): ΔR/R0 of MQ8, TGS2611, TGS2620 and SnO_2_Au-RGTO and the minimum of first derivative of MQ3, SnO_2_-RGTO and SnO_2_Au+Au-NW. These 7 inputs in k-NN algorithm were enough to obtain a classification rate of 94.7%. Misclassified samples belonged to Western Garda 1 (identified as Eastern Garda 2) and Eastern Garda 1 (confused with Western Garda 2 samples). It is evident that this is only a preliminary result that must be confirmed through further analysis, but this outcome seems encouraging.

### 2.2. Oil Chromatograms

The GC-MS data of the volatile organic compounds identified and quantified in the Garda PDO EVOOs showed the presence of 67 different key aroma compounds. The major categories of the compounds obtained from this study, including alcohols, aldehydes, carboxylic acids, esters, ketones, hydrocarbons, and terpenes, have been widely reported in the literature [14,15,16]. Out of 67, only 13 compounds were common to all the tested samples, and these were mainly acids (4 compounds) and alcohols (3 compounds), aldehydes (2 compounds) and ketones (2 compounds), as listed in Table 1. For each compound, the value of the peak area is reported as an arbitrary unit. Compounds were identified based on library matching with two libraries (NIST11 and FFNSC2).

Furthermore, there are 2 compounds present in all the EVO oils, but not in the olive oil sample. These two compounds are β-Ocimene and α-Farnesene. The former is a monoterpene found in a variety of plants and fruits that is found naturally as a mixture of isomers. The mixture, as well as the pure compounds, are oils with a pleasant odor. They are used in perfumery for their sweet herbal scent and are believed to act as plant defense and to have anti-fungal properties. The latter has been isolated from many plant sources and is a constituent of the natural coating of apples and pears and other fruit; it is used as a food flavoring oil.

Concerning the EVOOs tested on this work, it is possible to say that there are some characteristic compounds that identify the aromatic profile of the single oils. It was found that 3 compounds where present only in the non-PDO EVOOs:
Octane is a straight-chain alkane composed of 8 carbon atoms. It has a role as a xenobiotic. It is a colorless liquid with an odor of gasoline, less dense than water and insoluble in water;3,4-dimethyl-1,5-hexadiene is an olefin that has also been found in sugarcane bagasse oil [17];Methyl benzoate has a characteristic aroma that is sweet, creamy, anisic vanilla-like, slightly spicy, woody and powdery heliotropine-like. It has been found in the mushroom variety *Trametes graveolens*, feijoa fruit and peel (Feijoa sellowiana), white wine, cocoa, tea, guava, starfruit, Bourbon vanilla, Tahiti vanilla, mountain papaya, sapodilla fruit and Illicium verum [18]. 

From the sensor analysis, it was possible to identify the differences on the volatile profile of the four different Garda PDO EVOOs tested in this work. Even though these four oils share a common profile, there are other different volatile organic compounds that are found in only one of these oils. The data referring this information are presented in Table 2.

Silver decanoate appears to be the result of a violent reaction between silver powders and decanoic acid, a compound included on the list of the chemical substances that can be used as ingredients in an antimicrobial pesticide formulation applied to food-processing equipment and utensils [19]. Hence, it is plausible that silver decanoate could be obtained during the production process. On the contrary, bis(1-chloro-2-propyl) (3-chloro-1-propyl)phosphate is a pesticide that has also been found in marine salts samples coming from the Mediterranean Sea [20]. Although these compounds are not strictly characteristic of the aromatic profile of the two specific oils in question, they should be included in the list of compounds found exclusively in one of the analyzed samples since they are an index of the treatment of plants and the production process.

## 3. Materials and Methods

### 3.1. Sample Preparation and Experimental Design

A total of eight different olive oil samples were analyzed: four in the first phase (one olive oil, one Garda PDO EVOO and two non-PDO EVOOs) and four in the second phase (two Western Garda PDO EVOOs and two Eastern Garda PDO EVOOs). Regarding S3, in the first phase for each sample, 20 replicas were prepared for a total of 80 measurements; while in the second, 10 replicas were made. All samples were harvested in the years 2016/17, certified by the controlling body, and belonged to the two varieties from the west and east shores of Lake Garda.

The measurements were carried out in the autosampler HT2010H (HTA s.r.l., Brescia, Italy), which was connected to the S3 instrument. For the analysis, 5 mL of EVOO samples was enclosed in 20 mL chromatographic glass vials, sealed by a metal cap with PTFE-silicon membrane, crimped with an aluminum crimp, and placed randomly in the autosampler. Each vial was incubated at 35 °C for 5 min in the autosampler oven and shaken for 1 min during the incubation. The sample headspace was then drawn with an air flow of 50 sccm. The injection time (actual analysis time) was 2 min and the recovery time was 8 min. The analysis timeline (before, during, and after steps) and the method was set in the autosampler system prior to the analysis.

### 3.2. Small Sensor System (S3)

The state-of-the-art of S3 was completely designed, developed, and validated at the SENSOR Lab (University of Brescia, Brescia, Italy) in collaboration with Nano Sensor Systems (NASYS) s.r.l., a spin-off of the University of Brescia, Italy. It has been successfully applied in some selected applications [21,22,23,24,25,26,27]. The S3 system consists of a pneumatic part, an electronic part, an incubated sensor chamber housing an array of eight MOX gas sensors, and an online data acquisition and processing app. The flow sensors, temperature and humidity sensors, and actuators (valves and pumps) are all embedded inside the system. The multi-sensor array is connected with an autosampler HT2010H for the headspace sampling, supporting a 42-loading-site carousel. Table 3 shows MOX sensors details of S3 device; only sensors produced at SENSOR Lab are listed.

In each phase, commercial sensors were used, too. In phase 1, a TGS2611 was placed side-by-side with nanowires and RGTO sensors; in phase 2, five commercial sensors were used: MQ3, MQ8, MQ9, TGS2620 and TGS2611. In Figure 5, images of nanowires SnO_2_Au sensor are shown: on the left, there is a photo of the sensor with its gold support; on the right, a Scanning Electron Microscope Zeiss -LEO 1525 (Oberkochen, Germany) image where the nanowires can be seen.

### 3.3. SPME-GC-MS Analysis

The volatile components of the EVOOs were extracted and identified by the headspace GC-MS using the SPME method. It was used in parallel with the S3 analysis as a validation technique in order to characterize the headspace of the analyzed samples. A Shimadzu Gas Chromatograph GC2010 PLUS (Kyoto, KYT, Japan) interfaced with a single quadrupole mass spectrometer Shimadzu MS-QP2010 Ultra (Kyoto, KYT, Japan) and the HT280T autosampler were used for the analysis. The fiber used for the adsorption of the volatile compounds was DVB/CAR/PDMS-50/30 µm (Supelco Co. Bellefonte, PA, USA), which was exposed to the headspace of the vials after heating the samples in the autosampler oven at 35 °C for 20 min, with the aim of creating a headspace equilibrium. The desorption of the compounds took place in the GC-MS injector and the extracted compounds were separated using a capillary column MEGA-WAX (30 m × 0.25 mm × 0.25 μm, Agilent Technologies, Santa Clara, CA, USA). Ultrapure helium (99.99%) was used as a carrier gas, at a constant flow rate of 1.3 mL/min. After the collection of the mass spectra in the total ion current (TIC) mode, the identification of the volatile compounds was carried out using the NIST11 and the FFNSC2 mass spectra libraries.

### 3.4. Data Analysis Methods

Data from the sensors were analyzed with MATLAB 2015a software. Pre-processing of data consisted of an operation of normalization of the value of resistance respect to the first value recorded (R0). Hence, variation of normalized resistance (ΔR/R0) and minimum of first derivative were calculated. First derivative was calculated using finite differences, a method widely used to approximate numerical derivatives.

Principal Component Analysis (PCA) was performed in order to see how data could cluster and to reduce data dimensionality. This reduction was preliminary to k-Nearest Neighbors (k-NN) classification algorithm application. k-NN classifies samples by computing distances with nearest k points and then assigning them to the class more present in those k neighbors. Hence, classification rate was calculated as the ratio of exactly classified samples and total samples.

## 4. Conclusions

EVOO is a complex matrix and a beneficial food for human health. The aims of this work were to assess the capability of the system to distinguish between olive oil, non-PDO EVOOs and PDO EVOOs and discriminate four varieties of Garda PDO EVOOs, two from the eastern side of the lake, the others from the western one. This approach combining the S3 and SPME-GC-MS has shown good potential for evaluating and classifying different types of olive oils (virgin, extra virgin, PDO) and the geographical origins of the Garda DOP EVOOs. Olive oil analysis using the S3 is fast, easy, non-destructive, and automated, which helps to ensure the safety and authenticity of the product. With the GC-MS, the volatile compounds that characterize the samples under analysis were individuated. These results are compliant with those of the S3 analysis. Finally, the use of the S3 in parallel with the GC-MS enables more advanced, robust, efficient, and sensitive analytical methodologies focused on the analysis of the geographical origins of EVOOs and the main compounds from this complex matrix, while also ensuring the quality and traceability of the product.

## Figures and Tables

**Figure 1 molecules-24-01457-f001:**
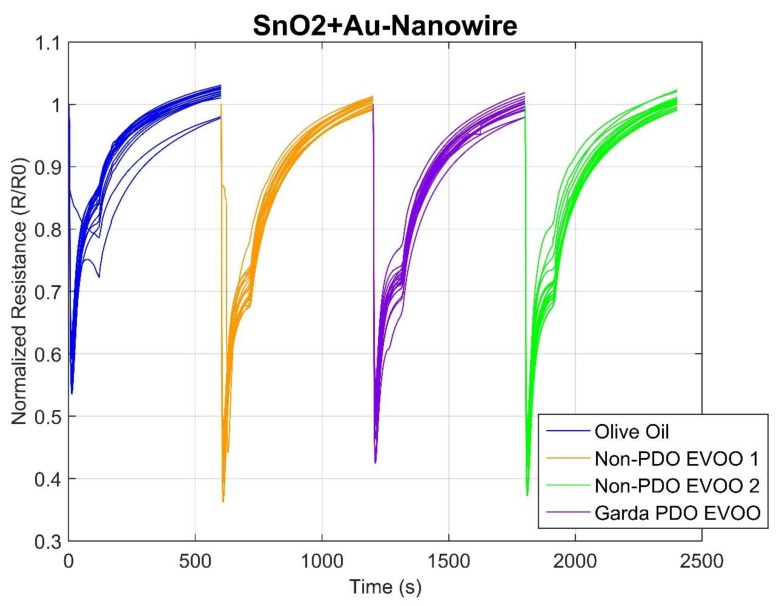
Typical resistance variation of a n-type sensor once exposed to oil volatile compounds. On the x-axis is time (s), on the y-axis is the normalized resistance.

**Figure 2 molecules-24-01457-f002:**
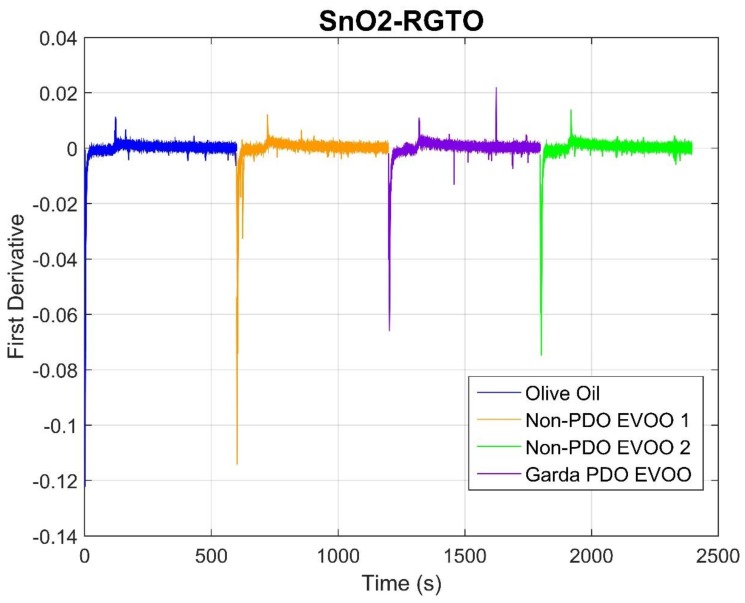
First derivative signals of a SnO_2_-RGTO sensor. On the x-axis is time (s), on the y-axis are first derivative values.

**Figure 3 molecules-24-01457-f003:**
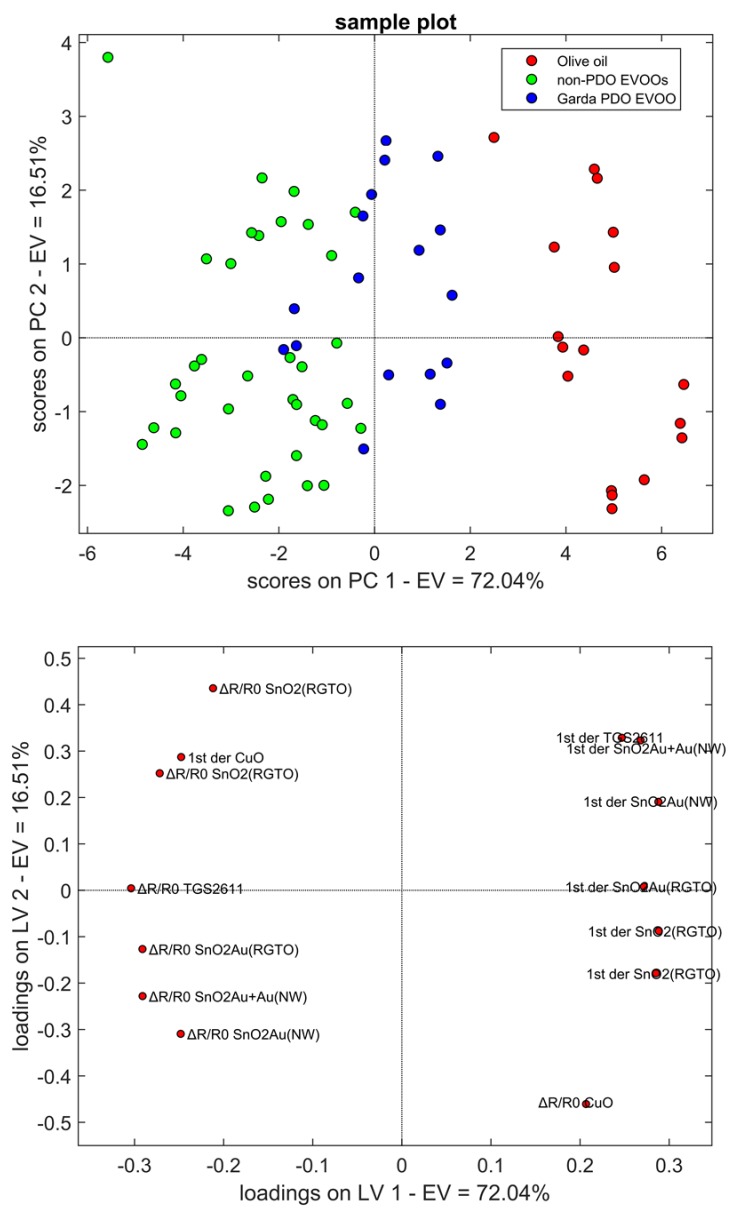
PCA score plot (top) and loading plot (bottom) of phase 1. PC1 variance = 72.04%, PC2 variance = 16.51%.

**Figure 4 molecules-24-01457-f004:**
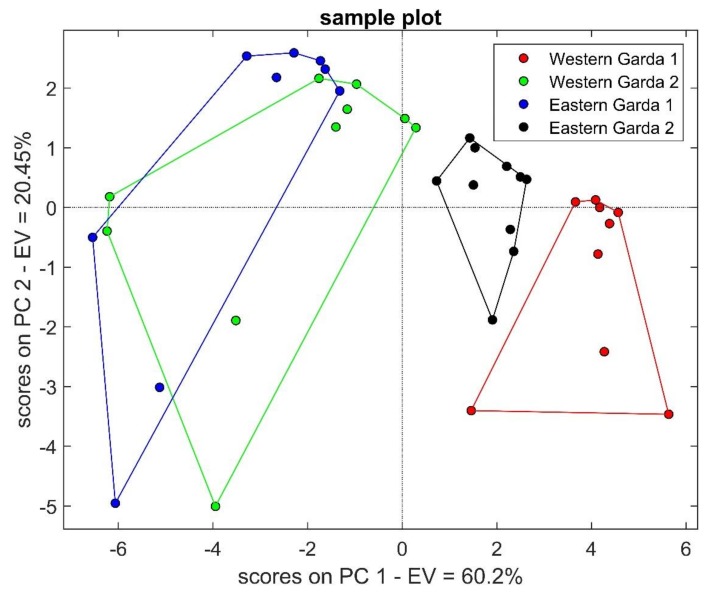
PCA scores plot of phase 2. PC1 variance = 60.2%, PC2 variance = 20.45%.

**Figure 5 molecules-24-01457-f005:**
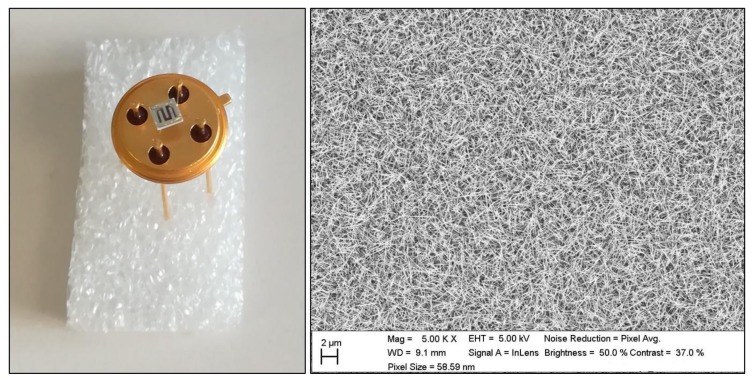
On the left, an image of a SnO_2_Au nanowire sensor with its support. On the right, a SEM image of the same sensor where the nanowires are visible.

**Table 1 molecules-24-01457-t001:** List of common compounds for all analyzed samples, with their retention time (min) and areas of the peaks (arbitrary unit).

Rt	Name	Phase 1				Phase 2			
		Garda PDO EVOO	Non-PDO EVOO 1	Non-PDO EVOO 2	Olive Oil	Western Garda EVOO 1	Western Garda EVOO 2	Eastern Garda EVOO 1	Eastern Garda EVOO 2
11.53	2-Hexenal	4.82 × 10^7^	4.46 × 10^7^	5.99 × 10^7^	1.81 × 10^6^	5.74 × 10^7^	3.32 × 10^7^	5.27 × 10^7^	4.71 × 10^7^
13.38	β-Ocimene	4.31 × 10^5^	6.25 × 10^5^	4.49 × 10^5^	0	1.17 × 10^6^	3.73 × 10^5^	6.21 × 10^5^	4.66 × 10^5^
14.57	Acetic acid, hexyl ester	1.96 × 10^5^	9.08 × 10^5^	2.63 × 10^5^	5.09 × 10^4^	3.62 × 10^5^	4.23 × 10^5^	2.76 × 10^5^	4.21 × 10^5^
17.03	3-Hexen-1-ol, acetate, (*Z*)-	5.66 × 10^5^	4.11 × 10^6^	8.24 × 10^5^	3.80 × 10^5^	1.18 × 10^6^	1.57 × 10^6^	1.08 × 10^6^	8.54 × 10^5^
19.20	1-Hexanol	1.94 × 10^6^	9.46 × 10^5^	2.35 × 10^6^	3.80 × 10^5^	1.47 × 10^6^	22.82 × 10^6^	2.90 × 10^6^	2.09 × 10^6^
20.98	3-Hexen-1-ol	5.03 × 10^5^	1.17 × 10^6^	1.56 × 10^6^	4.15 × 10^5^	1.22 × 10^6^	1.29 × 10^6^	1.07 × 10^6^	9.95 × 10^5^
21.29	Nonanal	6.26 × 10^5^	2.04 × 10^6^	9.31 × 10^5^	1.65 × 10^5^	8.68 × 10^5^	5.11 × 10^5^	5.77 × 10^5^	3.31 × 10^5^
22.41	2-Hexen-1-ol, (*E*)-	5.15 × 10^6^	1.13 × 10^6^	6.08 × 10^6^	5.72 × 10^5^	2.47 × 10^6^	4.44 × 10^6^	5.56 × 10^6^	3.00 × 10^6^
25.54	Ammonium acetate	1.12 × 10^6^	1.62 × 10^6^	2.44 × 10^6^	5.00 × 10^5^	1.07 × 10^6^	2.77 × 10^6^	3.36 × 10^6^	8.01 × 10^5^
35.99	Butanoic acid	2.45 × 10^5^	6.69 × 10^5^	1.29 × 10^6^	2.07 × 10^5^	4.01 × 10^5^	6.36 × 10^5^	4.55 × 10^6^	1.51 × 10^5^
42.37	α-Farnesene	2.46 × 10^5^	2.16 × 10^5^	3.37 × 10^5^	0	5.73 × 10^5^	1.97 × 10^5^	2.65 × 10^5^	1.51 × 10^5^
48.13	Hexanoic acid	7.88 × 10^5^	1.50 × 10^6^	2.13 × 10^6^	7.18 × 10^5^	6.30 × 10^5^	1.85 × 10^6^	5.77 × 10^6^	5.79 × 10^5^

**Table 2 molecules-24-01457-t002:** List of unique compounds for Garda PDO EVOOs, with their retention time (min) and areas of the peaks (arbitrary unit).

Rt	Name	Western Garda EVOO 1	Western Garda EVOO 2	Eastern Garda EVOO 1	Eastern Garda EVOO 2
2.40	4,4-Dimethyloxazolidine	0	0	0	2.12 × 10^4^
2.85	1-Fluorooctane	0	0	0	5.53 × 10^5^
4.17	4-Allyl-5-furan-2-yl-2,4-dihydro-[1,2,4]triazole-3-thione	0	0	0	1.43 × 10^4^
4.37	3,5-Dimethyloctane	0	0	0	2.56 × 10^5^
5.33	Heptylbenzene	0	0	0	3.39 × 10^4^
7.89	Cyclobut-1-enylmethanol	0	8.99 × 10^4^	0	0
10.52	trans-1,2-bis-(1-methylethenyl)cyclobutane	2.84 × 10^5^	0	0	0
14.08	1-(3-cyclohexen-1-yl)-ethanone	1.21 × 10^5^	0	0	0
18.12	Methyl heptenone	1.83 × 10^5^	0	0	0
20.04	4-Butoxy-1-butene	1.47 × 10^5^	0	0	0
24.52	2,6-Dimethyl-1,3,5,7-octatetraene, *E*,*E*-	1.45 × 10^5^	0	0	0
25.15	1-Octen-3-ol	0	5.46 × 10^4^	0	0
28.77	3,5-Octadien-2-one, (*E*,*E*)-	1.95 × 10^5^	0	0	0
30.80	1-Deoxy-d-arabitol	0	1.64 × 10^5^	0	0
64.37	*n*-Hexadecanoic acid	0	0	1.36 × 10^5^	0
69.21	Silver decanoate	0	0	1.03 × 10^5^	0
76.49	Cholest-7-en-3β,5α-diol-6α-benzoate	0	0	2.60 × 10^5^	0
81.33	tert-Butyldimethylsilyl 2-(2-(2-butoxyethoxy)ethoxy)acetate	1.20 × 10^5^	0	0	0
84.29	l-(+)-Ascorbic acid 2,6-dihexadecanoate	0	0	5.54 × 10^6^	0
86.46	Bis(1-chloro-2-propyl) (3-chloro-1-propyl)phosphate	1.84 × 10^5^	0	0	0
90.55	Octadecanoic acid	0	0	1.17 × 10^7^	0

**Table 3 molecules-24-01457-t003:** Type, composition, morphology and operating temperature for S3 sensors made at the SENSOR Laboratory. Sensors are divided for the two different phases of analysis.

Materials (Type)	Composition	Morphology	Operating Temperature (°C)
**Phase 1**			
SnO_2_Au (n)	SnO_2_ functionalized with Au clusters	RGTO	400 °C
SnO_2_ (n)	SnO_2_	RGTO	300 °C
SnO_2_ (n)	SnO_2_	RGTO	400 °C
SnO_2_Au + Au (n)	SnO_2_ grown with Au and functionalized with gold clusters	Nanowire	350 °C
SnO_2_Au (n)	SnO_2_ grown with Au	Nanowire	350 °C
CuO (p)	CuO	Nanowire	400 °C
**Phase 2**			
SnO_2_Au + Au (n)	SnO_2_ grown with Au and functionalized with gold clusters	Nanowire	350 °C
SnO_2_Au + Au (n)	SnO_2_ grown with Au and functionalized with gold clusters	Nanowire	400 °C
CuO (p)	CuO	Nanowire	350 °C
SnO_2_ (n)	SnO_2_	RGTO	450 °C
SnO_2_Au (n)	SnO_2_ with gold clusters	RGTO	400 °C

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
