# Peer review of "Multidisciplinary Approach to Characterizing the Fingerprint of Italian EVOO"

_molecules, 2019, doi:10.3390/molecules24081457_

Round 1
Reviewer 1 Report
1- Abstract – line 1 - insert (EVOO) after Extra virgin olive oil
2- Introduction – line 78 referring to gas sensors – a brief description is required about the sensors to clarify what compounds the sensors respond to, and if a series of compounds, these needs to be clarified.
3- Introduction – line 87 – referring to S3 sensors, clarification is required about the sensor as to what types of compounds the sensors are sensitive to?
4- Results and discussion:
5- Figure 1 – description is required about what ‘normalized resistance (R/R0)’ represents to guide the readers as how the sensors work? Description about x-axis (Time (s)) is required to help readers to understand the differences shown for the four samples. There is approximately 3-times difference in response time between the three PDO. How could this large difference be explained?
6- Figure 2 – description is required about what ‘first derivative’ represents to guide the readers about the data.
7- Table 1 – peak areas are typically reported as normalised and not as raw data. Further, large numbers are reported in scientific formats of x10, etc., so that to make it easier to distinguish differences between set of numbers.
8- In Tables 1 and 2 - RT (retention time), a unit is required and values reported to two decimal places would be sufficient.
9- A description is required for Tables 1 and 2 that compounds were identified based on library matching, and not by checking with standard reference compounds.
10- Materials and methods – the samples are from 2016/17 harvests, though the bottling production time (i.e. time typically stamped on bottles) for the eight olive oils samples is not reported.
Author Response
Dear Reviewer,
Thank you for your response and for your interest for this work. We enjoyed your comments, so we proceeded to make changes based on your suggestions.
1- Abstract – line 1 - insert (EVOO) after Extra virgin olive oil
EVOO inserted.
2- Introduction – line 78 referring to gas sensors – a brief description is required about the sensors to clarify what compounds the sensors respond to, and if a series of compounds, these needs to be clarified.
A brief description of gas sensors has been added.
3- Introduction – line 87 – referring to S3 sensors, clarification is required about the sensor as to what types of compounds the sensors are sensitive to?
They are non-specific sensors since they are MOX gas sensors, as we described in the brief description in lines 80-86. The real difference between our sensors and commercial ones is the production process that is described in Materials&Methods section.
4- Results and discussion:
5- Figure 1 – description is required about what ‘normalized resistance (R/R0)’ represents to guide the readers as how the sensors work? Description about x-axis (Time (s)) is required to help readers to understand the differences shown for the four samples. There is approximately 3-times difference in response time between the three PDO. How could this large difference be explained?
A more detailed description of the figure has been added. The normalization process and the x-axis have been explained.
6- Figure 2 – description is required about what ‘first derivative’ represents to guide the readers about the data.
The meaning of first derivative is explained at lines 126-131. A little description of the figure has been added.
7- Table 1 – peak areas are typically reported as normalised and not as raw data. Further, large numbers are reported in scientific formats of x10, etc., so that to make it easier to distinguish differences between set of numbers.
Numbers are reported in scientific format.
8- In Tables 1 and 2 - RT (retention time), a unit is required and values reported to two decimal places would be sufficient.
Rt unit has been added and two decimal places have been left.
9- A description is required for Tables 1 and 2 that compounds were identified based on library matching, and not by checking with standard reference compounds.
Description added (lines 193-194).
10- Materials and methods – the samples are from 2016/17 harvests, though the bottling production time (i.e. time typically stamped on bottles) for the eight olive oils samples is not reported.
Unfortunately, on the bottles that we used there is not a bottling time production but only the production lot.
Reviewer 2 Report
The text should contain references to figures. It should be completed wherever it is missing.
Retention indexes should be given instead of the retention times in Table 1.
Photos of metal oxide gas sensors would be desirable.
Author Response
Dear Reviewer,
Thank you for your response and your revision activity. We enjoyed your comments and proceeded to make changes based on your suggestions.
The text should contain references to figures. It should be completed wherever it is missing.
Missing references have been added.
Retention indexes should be given instead of the retention times in Table 1.
Thank you very much for reporting, but at this point in our work we are analyzing the compounds naturally present in the samples. In the future we plan to create a library that contains these substances and their RI-values.
Photos of metal oxide gas sensors would be desirable.
Figure 5 added; there are 2 images, one of the whole sensor and one of a SEM image of nanowires.
Reviewer 3 Report
The reviewed manuscript offers a possibility of distinguishing olive oil sample and determining their quality by using nano-sensors. This is an important applied chemistry topic – and I would like to see this manuscript published, eventually. However, at this point this manuscript is publishable only in a specialized journal rather than in a science journal like Molecules. The problem is that the sensors used are considered some kind of ‘black boxes’, with no actual relation to the molecules of analytes analyzed. The GC section does not help either because it does not relate the volatiles determined to the sensor response. If at least a cursory relation between GC and sensor responses were provided, the manuacript would be publishable in Molecules. Otherwise, I recommend the resubmission into a more applied, specialized journal.
Table 2 (on GC) needs substantial work. Not all the chemical names used are conventional, see my comments within the scanned file. N-Decanoic acid cannot elute between n-hexadecanoic and n-octadecanoic acids – there is a clear error in interpretation. A compound containing chorine and phosphorus appears to an impurity of a pesticide, so it should be marked as such. This part of the manuscript should be shown to an organic or analytical organic chemist.
There is one additional problem. Even though the manuscript is written in a relatively good English, it requires substantial editing. Missing words, odd word choices, incorrectly stated phrases, awkward sentences, etc, have to be addressed. I am providing some cursory editing as a starting point.
The title is too broad and does not represent the work conducted.

Author Response
Dear Reviewer,
Thank you for your response and for your interest for this work. We appreciated your comments and we proceeded to make changes based on your suggestions to improve the paper. We have followed most of your manual correction and we are very thakful for your grammar checks and corrections.
We checked compounds in table 2. In fact, you were right. N-hexanoic acid (molecular weight=256) appears at 64.37 min, while at 69.21 min there is Silver decanoate (molecular weight=278). Since the weight of the second compound is higher than the weight of the first compound, we think that now our interpretation is more adequate. Furthermore, we used conventional names where it was possible.
However, we disagree with your statement: “Otherwise, I recommend the resubmission into a more applied, specialized journal” since in the description of the special issue novel methods are appreciated (“We believe that this Special Issue will present challenging scientific approaches and recent and emerging issues and visions for future. Thus, it covers, but is not limited to, new methods and novel applications in food and beverages analysis”).
Round 2
Reviewer 3 Report
The reviewed manuscript has been improved; the authors have added some explanations on sensor functioning and edited the manuscript. This is an important applied chemistry topic – and I would like to see this manuscript published. I still contend, though, that the articles published in this journal should be based on molecules, i.e., chemistry, which is not the case when the sensors are used ‘black boxes’, with no connection to the specific analytes analyzed. However, I will leave this issue up to the Editor.
However, one part of the manuscript certainly must be corrected to avoid embarrassment. Namely, Table 2 (on GC) needs substantial work. In my previous review I pointed out that n-decanoic acid cannot elute between n-hexadecanoic and n-octadecanoic acids – there is a clear error in interpretation. The authors replaced n-decanoic acid with its silver salt – which is even more incorrect, for two reasons: silver cannot be present in the oil and, even if it were present, the silver salt would not elute from a GC column. On a similar token, the listed compound containing chorine and phosphorus appears to an impurity of a pesticide, so it should be marked as such. The authors ignored this comment of mine – but having this compound listed in Table 2 as an oil component is nonsense. This part of the manuscript must be shown (by the authors) to an expert in organic or analytical organic chemistry and reworked according to his/her recommendations – it simply cannot be published as is. If during this process a connection could be made between the sensors and GC, this adjustment would significantly upgrade the manuscript.
There is one more problem that must be addressed: The title is too broad and does not represent the work conducted. The word “valorization” is used incorrectly in the title. The title must be re-visited to reflect the novelty, i.e., the use of nano-sensors to “fingerprint” olive oil samples.
Author Response
Dear Reviewer,
Thank you again for your suggestions. Below, there are our point-to-point answers.
The reviewed manuscript has been improved; the authors have added some explanations on sensor functioning and edited the manuscript. This is an important applied chemistry topic – and I would like to see this manuscript published. I still contend, though, that the articles published in this journal should be based on molecules, i.e., chemistry, which is not the case when the sensors are used ‘black boxes’, with no connection to the specific analytes analyzed. However, I will leave this issue up to the Editor.
Response: I agree with you to leave the final decision to the Editor.
However, one part of the manuscript certainly must be corrected to avoid embarrassment. Namely, Table 2 (on GC) needs substantial work. In my previous review I pointed out that n-decanoic acid cannot elute between n-hexadecanoic and n-octadecanoic acids – there is a clear error in interpretation. The authors replaced n-decanoic acid with its silver salt – which is even more incorrect, for two reasons: silver cannot be present in the oil and, even if it were present, the silver salt would not elute from a GC column. On a similar token, the listed compound containing chorine and phosphorus appears to an impurity of a pesticide, so it should be marked as such. The authors ignored this comment of mine – but having this compound listed in Table 2 as an oil component is nonsense. This part of the manuscript must be shown (by the authors) to an expert in organic or analytical organic chemistry and reworked according to his/her recommendations – it simply cannot be published as is. If during this process a connection could be made between the sensors and GC, this adjustment would significantly upgrade the manuscript.
Response: Regarding the two cited compounds, we made deeper researches to find a plausible explanation.
In a book entitled “Sittig's Handbook of Pesticides and Agricultural Chemicals” (2nd Edition), we found that decanoic acid is “used in cleaning, sanitizing and disinfecting applications for food processors”. Decanoic acid is included on the list of the chemical substances that can be used as ingredients in an antimicrobial pesticide formulation applied to food-processing equipment and utensils. Moreover, decanoic acid “reacts violently with […] silver powders or dust”. Finally, we think that it is not so incorrect. “Bis(1-chloro-2-propyl)(3-chloro-1-propyl)phosphate” is a pesticide. We added a paragraph below Table 2 to explain their presence in the table.
There is one more problem that must be addressed: The title is too broad and does not represent the work conducted. The word “valorization” is used incorrectly in the title. The title must be re-visited to reflect the novelty, i.e., the use of nano-sensors to “fingerprint” olive oil samples.
Response: We changed the title from “Multidisciplinary approach to valorize Italian EVOO” to “Multidisciplinary approach to characterize the fingerprint of Italian EVOO”.